# VFRTok: Variable Frame Rates Video Tokenizer with Duration-Proportional Information Assumption

**Tianxiong Zhong**[1][*], **Xingye Tian**[2], **Boyuan Jiang**[2], **Xuebo Wang**[2][†],
**Xin Tao**[2], **Pengfei Wan**[2], **Zhiwei Zhang**[1][†]

[1]Beijing Institute of Technology, [2]Kling Team, Kuaishou Technology

inkosizhong@gmail.com, {tianxingye,jiangboyuan,wangxuebo}@kuaishou.com,
{taoxin,wanpengfei}@kuaishou.com, zwzhang@bit.edu.cn,

## Abstract

Modern video generation frameworks based on Latent Diffusion Models suffer from inefficiencies in tokenization due to the Frame-Proportional Information Assumption. Existing tokenizers provide fixed temporal compression rates, causing the computational cost of the diffusion model to scale linearly with the frame rate. The paper proposes the Duration-Proportional Information Assumption: the upper bound on the information capacity of a video is proportional to the duration rather than the number of frames. Based on this insight, the paper introduces VFRTok, a Transformer-based video tokenizer, that enables variable frame rate encoding and decoding through asymmetric frame rate training between the encoder and decoder. Furthermore, the paper proposes Partial Rotary Position Embeddings (RoPE) to decouple position and content modeling, which groups correlated patches into unified tokens. The Partial RoPE effectively improves content-awareness, enhancing the video generation capability. Benefiting from the compact and continuous spatio-temporal representation, VFRTok achieves competitive reconstruction quality and state-of-the-art generation fidelity while using only $1/8$ tokens compared to existing tokenizers. The code and weights are released at: https://github.com/KwaiVGI/VFRTok.

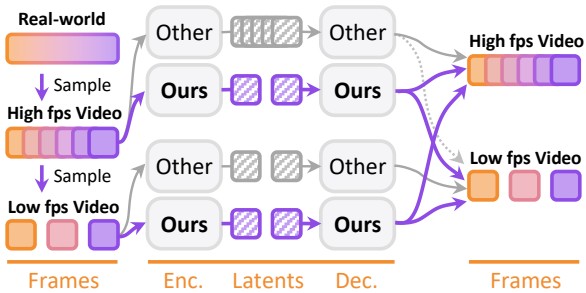

Figure 1: VFRTok is based on the Duration-Proportional Information Assumption. The number of tokens for other tokenizers grows with frame rate. VFRTok maintains a fixed length latents tied to video duration and supports asymmetric frame-rate encoding and decoding.

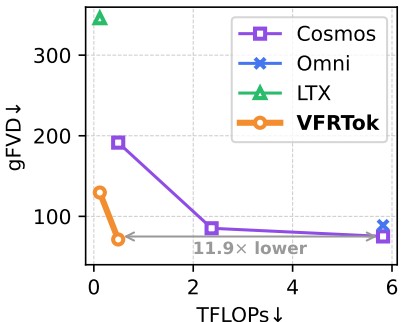

Figure 2: Efficiency-quality trade-off, where lower-left indicates better performance. VFRTok provides a more efficient latent representation.

---

[*]This work was conducted during the author's internship at Kling Team.
[†]Corresponding Authors.

39th Conference on Neural Information Processing Systems (NeurIPS 2025).

# 1 Introduction

Recently, Latent Diffusion Model (LDM) is widely used in image [2, 21, 24, 25, 36, 38, 39] and video generation [1, 18, 22, 35, 41], comprising two main components: a tokenizer and a diffusion model. The tokenizer compresses data from the original high-dimensional pixel space to a low-dimensional latent space, which reduces the training and inference overhead of the Diffusion Transformers (DiT) by a quadratic factor. The video tokenizers [1, 11, 18, 33, 35, 41] eliminate intra- and inter-frame redundancy in the video by simultaneously compressing both temporal and spatial dimensions.

Existing video tokenizers [1, 11, 18, 33, 35, 41] are built upon the Frame-Proportional Information Assumption, which assumes a fixed compression rate for a given number of video frames (Figure 1). These tokenizers are trained on and designed to generate videos with a fixed frame rate. High frame rate videos require a larger number of tokens for representation, resulting in the number of tokens increase linearly with the frame rate, which significantly increases the computational overhead.

Video is the result of continuous space-time being sampled uniformly. The amount of observable information in continuous space-time serves as the natural upper bound on the information contained in the video. Intuitively, when the video frame rate increases from 12 frame per second (FPS) to 24 FPS, the change can be clearly observed, whereas the difference between 60 FPS to 120 FPS yields more subtle changes. When a camera samples a motion trajectory $x(t)$ at a sampling frequency $f_s$, the resulting discrete samples can be used to estimate the continuous trajectory. An interpolation algorithm is typically employed to reconstruct the continuous motion trajectory $\hat{x}(t)$ from these discrete observations. According to interpolation error estimation theory in numerical analysis, the upper bound of the estimation error $E_{max}$ is related to $f_s$ as follows:

$$E_{max} = \sup_t |x(t) - \hat{x}(t)| \leq \frac{C \cdot \sup_t |x^{(k)}(t)|}{f_s^k}. \tag{1}$$

where $k$ represents the order of accuracy of the interpolation algorithm, $\sup_t |x^{(k)}(t)|$ is an upper bound on the $k^{th}$ derivative of the true trajectory $x(t)$, and $C$ is a positive constant. Equation (1) implies that the information gain diminishes as the frame rate increases.

Motivated by this insight, we propose the Duration-Proportional Information Assumption, which guides the design of compression rates that scale with video duration (Figure 1). Specifically, we introduce a **V**ariable **F**rame **R**ates video **Tok**enizer, VFRTok, which is a query-based Transformer tokenizer and enables the encoder and decoder to process different frame rates. VFRTok uses asymmetric frame rate training between the encoder and decoder to learn continuous spatio-temporal representations, that enables the generation of videos at arbitrary frame rates.

Furthermore, we observed that both existing tokenizers [1, 11, 33] and VFRTok exhibit a grid-based pattern, where each latent token tends to attend to a fixed spatial location within the temporal sequence. To achieve a more compact representation, we aim to strengthen VFRTok's content-awareness. Our analysis illustrates that the latent representations in VFRTok are strongly influenced by the positional prior introduced by Rotary Position Embeddings (RoPE), which hinders effective content modeling. Therefore, we propose Partial RoPE, which applies RoPE only to a subset of the attention heads, encouraging a functional separation between positional and content encoding. Experimental results show that the Partial RoPE effectively enhances the model's capacity for content modeling.

Benefiting from these designs, VFRTok efficiently reconstructs and generates videos. For example, as shown in Figure 2, VFRTok achieves better generation results than existing tokenizers [1, 33] while requiring $11.9\times$ less computation overhead. Meanwhile, VFRTok natively supports video frame interpolation, enabling frame rates to be increased from 12 FPS to 120 FPS. In summary, we highlight the main contributions:

1. We propose the Duration-Proportional Information Assumption and the first high-compression video tokenizer with variable frame rate.

2. We introduce Partial RoPE to mitigate the influence of video patch position priors on latent tokens and enhance content-awareness.

3. Experiments show that we can achieve comparable reconstruction and state-of-the-art generation while using only $1/8$ tokens compared to existing tokenizers.

## 2 Related Work

### 2.1 Video Tokenizer

Video data consists of a series of continuously changing frames and video tokenizers compress the video in both temporal and spatial dimensions. Therefore, video generation task must consider inter-frame consistency to avoid problems such as flickering and jittering. Early LDM [22] directly uses image tokenizers in a frame-by-frame compression pattern. In contrast, modern video tokenizers [1, 11, 16, 18, 31, 33, 35, 41] generally use 3D computation modules that include the temporal dimension.

Mainstream video tokenizers generally provide a compression rate of $4 \times 8 \times 8$ [16, 18, 33, 35, 41], which meaning a 16-frame video with $256 \times 256$ resolution can be compressed into $4 \times 32 \times 32 = 4096$ tokens. A few works propose tokenizers with higher compression rates [1, 11, 20, 31]. For instance, Cosmos Tokenizer [1] leverages wavelet transform and provides a series of tokenizers with compression rates from $4 \times 8 \times 8$ to $8 \times 16 \times 16$. LTX-VAE [11] cascades additional downsample layers and achieves a compression rate of up to $8 \times 32 \times 32$ through a spatial-to-depth approach.

However, existing video tokenizers [1, 11, 16, 18, 31, 33, 35, 41] rely on the Frame-Proportional Information Assumption and process sequences with fixed frame rates. Different from these tokenizers, which model discrete video frame pixel space, VFRTok models continuous spatio-temporal information using asymmetric frame rate training strategy between the encoder and decoder.

### 2.2 Query-based Visual Tokenizer

Unlike traditional grid-based tokenizers [2, 24, 25], query-based image tokenizers [5, 6, 37] encode images into compact 1D latent representations. These tokenizers employ a Transformer-based framework and use learnable latent tokens to query information from the image patches. For instance, TiTok [37] proposes a VQ-based tokenizer and represent an image with 32 tokens for reconstruction and MaskGiT [4]-style generation. SoftVQ-VAE [5] leverages soft categorical posteriors to increase the representation capacity of the latent space, which can be applied to both autoregressive- and diffusion-based image generation. MAETok [6] further improves the diffusibility [6, 36] of vanilla AutoEncoder (AE) by regularizing it with mask modeling and auxiliary decoders. LARP [32] extends the query-based structure to video tokenizer, adapt to autoregressive video generative models.

The query-based image tokenizers [5, 6, 37] achieve comparable or better performance than the grid-based image tokenizers [2, 24, 25], using significantly fewer tokens. This shows that the query-based structure has the potential to better eliminate redundant information. More importantly, it provides the possibility to encode variable-length data into fixed-length latent representations.

## 3 Method

### 3.1 Architecture

Given a video $X \in \mathbb{R}^{F \times H \times W \times 3}$, we want to obtain its compressed representation $Z = \mathcal{E}(X) \in \mathbb{R}^{N \times d}$, where $N$ represents the number of tokens, and the reconstructed video $\hat{X} = \mathcal{D}(Z) \in \mathbb{R}^{F \times H \times W \times 3}$. Figure 3 illustrates the architecture of VFRTok. We adopt the AE architecture and use ViT [9] as the backbone for both the encoder and decoder. On the encoder side, the input video $X$ is patchified into a serials of spatial-temporal patch tokens $x \in \mathbb{R}^{\frac{F}{p_F} \times \frac{H}{p_H} \times \frac{W}{p_W} \times h}$, where $p_F \times p_H \times p_W$ is the patch size and $h$ is the hidden dimension. Similarly, the decoder recovers the reconstructed patch tokens $\hat{x} \in \mathbb{R}^{\frac{F}{p_F} \times \frac{H}{p_H} \times \frac{W}{p_W} \times h}$ back to the pixel-space $\hat{X}$.

VFRTok is based on the Duration-Proportional Information Assumption, which requires a continuous spatio-temporal representation. We adopt a query-based approach [5, 6, 37] that encodes video by querying grid-based patch tokens with fixed-length latent tokens, and decodes by reversing this process, where patch tokens query latent representations. To align equal-duration videos of different frame rates, we modify the temporal modeling in 3D RoPE [19, 28, 34] from frame-index-based position encoding [33] to a timestamp-based approach, detailed in Section 3.2. Furthermore, as described in Section 3.3, we propose an improved RoPE implementation, Partial RoPE, which applies RoPE to a subset of attention heads to decouple positional information from content modeling.

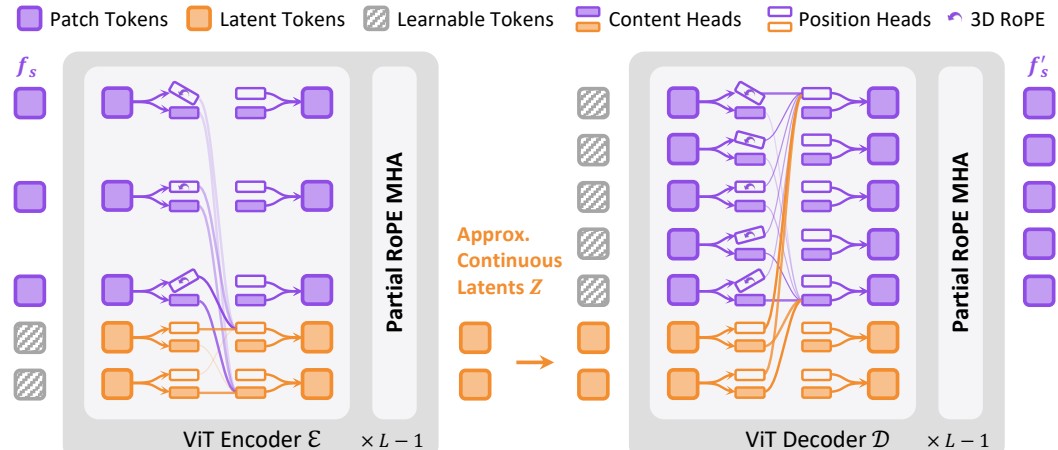

Figure 3: VFRTok adopts a query-based ViT architecture. VFRTok models variable-length patch tokens with fixed-length latent tokens, to support the encoding and decoding of variable frame rate videos. VFRTok further introduces Partial RoPE, which applies RoPE to a subset of attention heads to decouple positional information from content modeling.

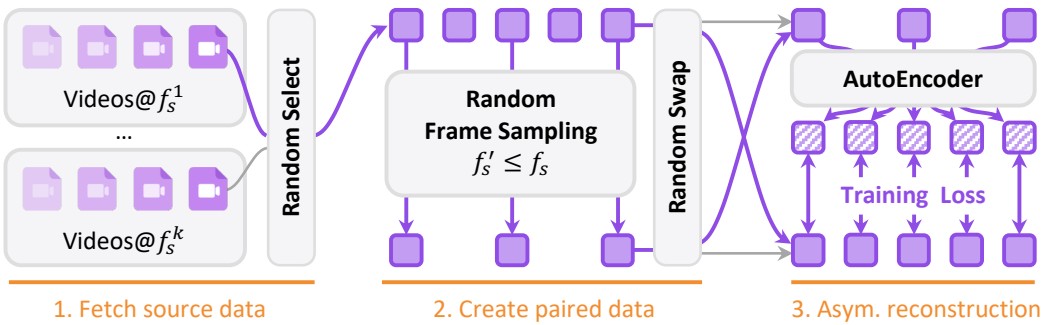

Figure 4: Asymmetric frame rate training strategy between encoder and decoder. We construct paired high and low frame rate videos to learn continuous spatio-temporal representations.

## 3.2 Duration-Proportional Compression

VFRTok accepts data of any frame rate but with equal duration, learns the continuous spatio-temporal information it represents, and encodes this information into fixed-length latent representations. Specifically, it concatenates learnable latent tokens $z \in \mathbb{R}^{N \times h}$ with the input $x$ before ViT encoding. The patch tokens are dropped at the bottleneck of the encoder, and only the latent tokens are retained and mapped to the low-dimension compressed representation $Z$. On the decoder side, $Z$ is first remapped to the hidden feature $\hat{z}$, and concatenated with the learnable patch tokens $\hat{x}$. As shown in Figure 3, based on the flexible architecture, VFRTok allows the encoder and decoder to apply different frame rates. Since they represent data of the same duration, videos with higher FPS contain more frames, which in turn translates into more video patches.

To achieve Duration-Proportional compression, we use 3D RoPE [28] to model the positional dependencies among patch tokens and replace the frame-driven rotation angle to a timestamp-driven formulation. Specifically, we first generalize RoPE [28] to video data by splitting the channels in each attention head into three parts, which are used to encode the positional information along the temporal, vertical, and horizontal spatial axis. Given a patch token at temporal-spatial position $(t, j, k)$ in a $F \times H \times W$ patch token space, the rotation matrix $R_{t,j,k} \in \mathbb{R}^{n \times n}$ can be represented as:

$$R_{t,j,k} = \begin{pmatrix} R_t^T & \mathbf{0} & \mathbf{0} \\ \mathbf{0} & R_j^H & \mathbf{0} \\ \mathbf{0} & \mathbf{0} & R_k^W \end{pmatrix}, R_i^* = \begin{pmatrix} R_{i,1}^* & \mathbf{0} & \mathbf{0} \\ \mathbf{0} & \ddots & \mathbf{0} \\ \mathbf{0} & \mathbf{0} & R_{i,\frac{n}{6}}^* \end{pmatrix}, R_{i,c}^* = \begin{pmatrix} \cos\theta_{i,c}^* & -\sin\theta_{i,c}^* \\ \sin\theta_{i,c}^* & \cos\theta_{i,c}^* \end{pmatrix}, \quad (2)$$

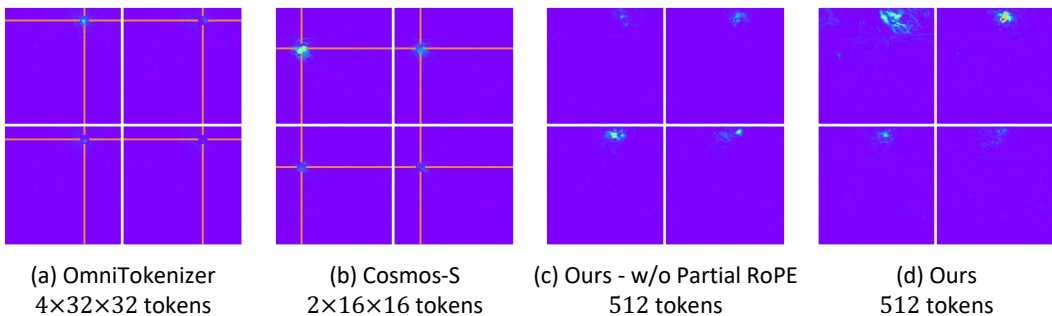

(a) OmniTokenizer
4×32×32 tokens

(b) Cosmos-S
2×16×16 tokens

(c) Ours - w/o Partial RoPE
512 tokens

(d) Ours
512 tokens

Figure 5: Visualization of the region affected by a single token. The heat map is an overlay of 100 samples, showing the 1st, 5th, 9th, and 13th frames for each method. The reference lines are drawn in the grid-based approaches to indicate the spatial position of the token within the grid.

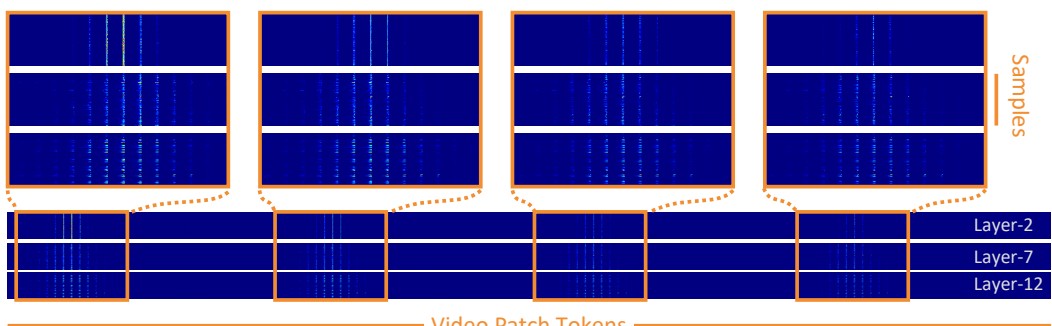

Figure 6: The latent-to-patch attention map from different Transformer layers reveals the interference of position prior on content modeling. Each row represents the intensity of information flow across patch tokens, and rows correspond to different samples.

where $n$ represents the number of channels of an attention head, $\theta_{i,c}$ is the rotation angle of the $c$-th channel of the $i$-th token. Specifically, as shown in Equation (3), VFRTok directly uses the spatial index $\{j, k\}$ of the patch to calculate the rotation angle $\{\theta_{j,c}^H, \theta_{k,c}^W\}$, and converts the temporal index $t$ to a timestamp $t/f_s$ to compute the corresponding rotation angles $\theta_{t,c}^F$.

$$\theta_{t,c}^F = C \times \frac{t}{f_s} \times 10000^{-\frac{6c}{n}}, \ \theta_{j,c}^H = j \times 10000^{-\frac{6c}{n}}, \ \theta_{k,c}^W = k \times 10000^{-\frac{6c}{n}}, \tag{3}$$

where $f_s$ is the frame rate and $C$ is an optional normalization coefficient. This makes videos of the equal duration share the same maximum rotation angle. Meanwhile, as the frame rate increases, the angular difference between adjacent frames decreases.

As shown in Figure 4, we employ the encoder-decoder asymmetric frame rate training strategy to guide the latents in encoding continuous spatio-temporal information. Firstly, we divide the dataset into multiple buckets according to their original frame rate. For each batch, we randomly fetch data with frame rate $f_s$ from a bucket. Secondly, we randomly select a downsampling factor $\tau \leq 1$ from a predefined range and extract frames with $f_s' = f_s \cdot \tau$ from the video. The two video sequences are randomly swapped to determine the frame rate of VRFTok's encoder $f_s^{\mathcal{E}}$ and decoder $f_s^{\mathcal{D}}$. Finally, it encodes the video sequence with $f_s^{\mathcal{E}}$ and learns to reconstruct the video with $f_s^{\mathcal{D}}$. The configurations vary across GPUs, enabling VFRTok to learn more continuous spatio-temporal representations.

### 3.3 Partial RoPE

To analyze what information in the video is compressed by each token, we use PCA [14] to ablate latent tokens individually, retain the remaining tokens for reconstruction, and observe the degradation position of the reconstructed video.

$$\Delta X_{\setminus i} = \left| \mathcal{D}(Z) - \mathcal{D}(Z_{\setminus i}) \right|, \ Z_{\setminus i} = \left( Z_0, ..., Z_{i-1}, \mathrm{PCA}_i^{-1}(\mathbf{0}), Z_{i+1}, ..., Z_N \right), \tag{4}$$

where $\mathrm{PCA}_i^{-1}(\cdot)$ is the inverse PCA based on the coefficient corresponding to the $i$-th token $Z_i$ across on a batch of samples, and $|\cdot|$ is an element-wise absolute. As shown in Figure 5(a,b,c), the latents of existing methods [1, 33] and the vanilla query-based approach exhibits a regular grid property. The token corresponds to pixels at a relatively fixed position across different frames. Existing methods [1, 33] are based on grid designs, and their tokens strictly follow the guidance of the grid. In contrast, our query-based approach is expected to provide more content-aware latent expressions. Therefore, we investigate the underlying cause.

We visualize the latent-to-patch attention map of the decoder across all layers and heads. Formally, for a given attention map $A \in \mathbb{R}^{(M+N)\times(M+N)}$, where $M = \frac{F}{p_F} \times \frac{H}{p_H} \times \frac{W}{p_W}$ is the number of patch tokens, it can be considered as a block matrix:

$$A = \begin{pmatrix} A_{\mathrm{patch}} \in \mathbb{R}^{M\times M} & A_{\mathrm{latent}\to\mathrm{patch}} \in \mathbb{R}^{M\times N} \\ A_{\mathrm{patch}\to\mathrm{latent}} \in \mathbb{R}^{N\times M} & A_{\mathrm{latent}} \in \mathbb{R}^{N\times N} \end{pmatrix}. \tag{5}$$

For a given sample, we extract each column from $A_{\mathrm{latent}\to\mathrm{patch}}$, which represents the intensity of information flow across patch tokens. We concatenate together the arrays of multiple samples to form Figure 6, where the rows correspond to samples and the columns correspond to video patch tokens. Figure 6 reveals that the attention is dominated by positional prior. Specifically, the attention distribution shows a long-short pattern across different samples, where the long periodicity arises from temporal priors, while short periodicity originates from spatial priors.

To alleviate this problem, we propose Partial RoPE, which divides the attention heads into position heads and content heads. For the position heads, we adopt the original 3D RoPE, while for the content heads, we simply remove the RoPE to adequately learn content information. We use $\tau_{\mathrm{RoPE}}$ to control the proportion of position heads. The experimental results demonstrate that Partial RoPE effectively enhances the content-awareness and the generation quality of VFRTok.

## 4 Experiments

### 4.1 Setup

**Training details.** We train VFRTok $\phi$ using the standard reconstruction objective.

$$\mathcal{L} = \mathcal{L}_{\mathrm{recon}} + \lambda_1 \mathcal{L}_{\mathrm{percept}} + \lambda_2 \cdot \lambda_\nabla \mathcal{L}_{\mathrm{adv}}, \ \lambda_\nabla = \frac{\left\|\nabla_\phi\left(\mathcal{L}_{\mathrm{recon}} + \lambda_1 \mathcal{L}_{\mathrm{percept}}\right)\right\|}{\left\|\nabla_\phi \mathcal{L}_{\mathrm{adv}}\right\|}, \tag{6}$$

where $\mathcal{L}_{\mathrm{recon}}$, $\mathcal{L}_{\mathrm{percept}}$, and $\mathcal{L}_{\mathrm{adv}}$ are the L1 reconstruction loss, perceptual loss [13, 17], and adversarial loss [10], respectively, and $\lambda_\nabla$ represents adaptive weight. The hyperparameters are set to $\lambda_1 = 1$ and $\lambda_2 = 0.2$. The patch size of VFRTok is set to $p_F \times p_H \times p_W = 4 \times 8 \times 8$. The Partial RoPE ratio is set to $\tau_{\mathrm{RoPE}} = 0.5$, indicating that 6 of the 12 attention heads employ RoPE. An implicit advantage of VFRTok is that the number of tokens $N$ and channels $d$ can be easily adjusted. To balance generation quality and training efficiency, we provide VFRTok-L and VFRTok-S with the same latent capacity but differ in token count, $Z^L \in \mathbb{R}^{512\times32}$ and $Z^S \in \mathbb{R}^{128\times128}$. To evaluate the tokenizers, we modified LightningDiT-XL/1 [36] to support video generation. If not specified, the models process video with 2/3s duration, which is 16 frames at $f_s = 24$. For fair comparison, the first frame of the image and video joint tokenizers [1, 11, 33] decoded using the image token, is not counted in the metrics, nor counted in the model calculation costs.

**Datasets.** VFRTok is trained in a three-stage manner. First, it is initialized on the ImageNet-1K [8] for 30,000 steps with a batch size of 512. Then, it is pre-trained on the K600 dataset [3] for 200,000 steps with a batch size of 64, employing asymmetric FPS training $f_s^{\mathcal{E}} = f_s^{\mathcal{D}} \in \{12, 18, 24, 30\}$. Finally, 22 sequences of 120 FPS data from the BVI-HFR dataset [7] are added for 100,000 steps with a batch size of 16, using FPS settings $f_s^{\mathcal{E}}, f_s^{\mathcal{D}} \in \{12 + 6k \mid k = 0, 1, \ldots, 18\}$. Reconstruction evaluation is performed on the K600 [3] validation set and the UCF101 [26] dataset. VFRTok-L and VFRTok-S are trained on a single node equipped with 8 H800 GPUs, requiring 4 days, respectively. The DiT models [36] are trained and evaluated on the K600 [3] and UCF101 [26] datasets, respectively, using label-based Classifier-Free Guidance [12] (CFG). To demonstrate the advantages of VFRTok in high frame rate video generation, the DiT model was also trained on 60 FPS data from the LAVIB [27] dataset. DiTs [36] are trained for 100,000 steps with a batch size of 128 on UCF101 [26] and K600 [3] datasets, and 50,000 steps with a batch size of 48 on LAVID [27] dataset, respectively. The

Table 1: Comparison of reconstruction performance across multiple datasets for different tokenizers. Gray highlights indicate cases where VFRTok is superior or comparable.

| Method | #Tokens | #Dim. | K600 | | | | UCF101 | | | |
| --- | --- | --- | --- | --- | --- | --- | --- | --- | --- | --- |
| | | | PSNR↑ | SSIM↑ | LPIPS↓ | rFVD↓ | PSNR↑ | SSIM↑ | LPIPS↓ | rFVD↓ |
| Omni [33] | 4096 | 8 | 29.35 | 0.9143 | 0.0573 | 5.14 | 28.95 | 0.9239 | 0.0505 | 10.21 |
| Cosmos-L [1] | 4096 | 16 | 33.34 | 0.9284 | 0.0546 | 3.28 | 33.42 | 0.9372 | 0.0439 | 5.55 |
| Cosmos-M [1] | 2048 | 16 | 31.66 | 0.9068 | 0.0710 | 6.77 | 31.70 | 0.9177 | 0.0575 | 13.67 |
| Cosmos-S [1] | 512 | 16 | 28.46 | 0.8445 | 0.1209 | 70.26 | 28.26 | 0.8577 | 0.1046 | 104.51 |
| LTX [11] | 128 | 128 | 32.04 | 0.9100 | 0.0582 | 22.11 | 32.02 | 0.9202 | 0.0508 | 35.32 |
| VFRTok-L | 512 | 32 | 31.63 | 0.9104 | 0.0394 | 4.64 | 31.54 | 0.9193 | 0.0391 | 13.79 |
| VFRTok-S | 128 | 128 | 31.55 | 0.9089 | 0.0401 | 6.02 | 31.50 | 0.9178 | 0.0401 | 15.55 |

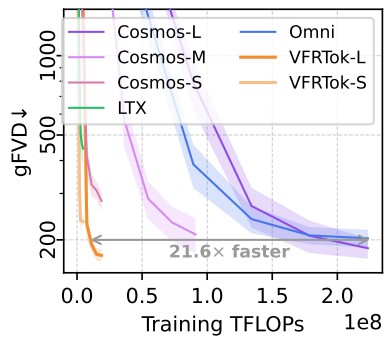

Figure 7: Convergence speed of different tokenizers on UCF101.

Table 2: Comparison of unconditional and CFG video generation in terms of gFVD↓ and TFLOPs↓. Best results are **bolded**; gray highlights indicate VFRTok-S superiority.

| Method | TFLOPs | K600 | | UCF101 | |
| --- | --- | --- | --- | --- | --- |
| | | w/o CFG | w/ CFG | w/o CFG | w/ CFG |
| Omni [33] | 5.82 | 521.89 | 242.54 | 480.60 | 88.89 |
| Cosmos-L [1] | 5.82 | 620.07 | 302.58 | 476.08 | 75.11 |
| Cosmos-M [1] | 2.37 | 554.95 | 125.02 | 497.01 | 85.22 |
| Cosmos-S [1] | 0.49 | 569.58 | 210.21 | 678.37 | 191.49 |
| LTX [11] | 0.12 | 615.28 | 358.48 | 735.38 | 345.82 |
| VFRTok-L | 0.49 | **323.37** | **124.78** | **377.50** | **71.34** |
| VFRTok-S | 0.12 | 412.97 | 131.34 | 443.41 | 129.55 |

training cost correlates with the number of latent tokens, ranging from 5 hours using 8 H800 GPUs (VFRTok-S) to 3 days using 16 H800 GPUs (Cosmos-L [1]).

**Metrics.** For the reconstruction task, we use PSNR, SSIM, and LPIPS [40] to perform frame-wise evaluation. Meanwhile, we use reconstruction FVD [30] (rFVD) as a spatio-temporal metric. For the generation task, we use generation FVD [30] (gFVD) to evaluate frame quality with and without CFG [12]. We use floating-point operations (TFLOPs) to evaluate the calculation costs of all DiTs.

## 4.2 Video Reconstruction

Table 1 reports reconstruction metrics of various methods [1, 11, 33] across different datasets [3, 26]. It shows that all models in the VFRTok family share similar reconstruction quality, attributed to their shared latent token capacity. VFRTok achieves comparable quality with Cosmos-M [1] and OmniTokenizer [33], using only $1/4$ latent capacity. In comparisone, LTX-VAE [11] performs better than VFRTok on PSNR, but VFRTok achieves significantly better LPIPS [40] and rFVD [30]. It is worth noting that Cosmos [1] and LTX-VAE [11] use larger and licensed training dataset, so VFRTok still has the potential to achieve better results.

## 4.3 Video Generation

As shown in Table 2, VFRTok-L achieves state-of-the-art (SOTA) performance in the unconditional and CFG [12] generation on UCF101 [26] and K600 [3] dataset. The overhead of VFRTok-L is only $8.4\%$ of OmniTokenizer and Cosmos-L, and $20.6\%$ of Cosmos-M. Furthermore, we use gray highlighting to indicate cases where VFRTok-S is superior. For example, VFRTok-S consistently outperforms LTX-VAE [11] and Cosmos-S [1] in all tasks. We show the convergence speed of different methods on the UCF101 [26] dataset in Figure 7, where the error bars reflect different CFG scales $\{2, 3, 4\}$. Due to the high computational cost of DiT [36] when using existing tokenizers [1, 33], we randomly conduct a fair comparison on 1,000 samples. The experimental result shows that

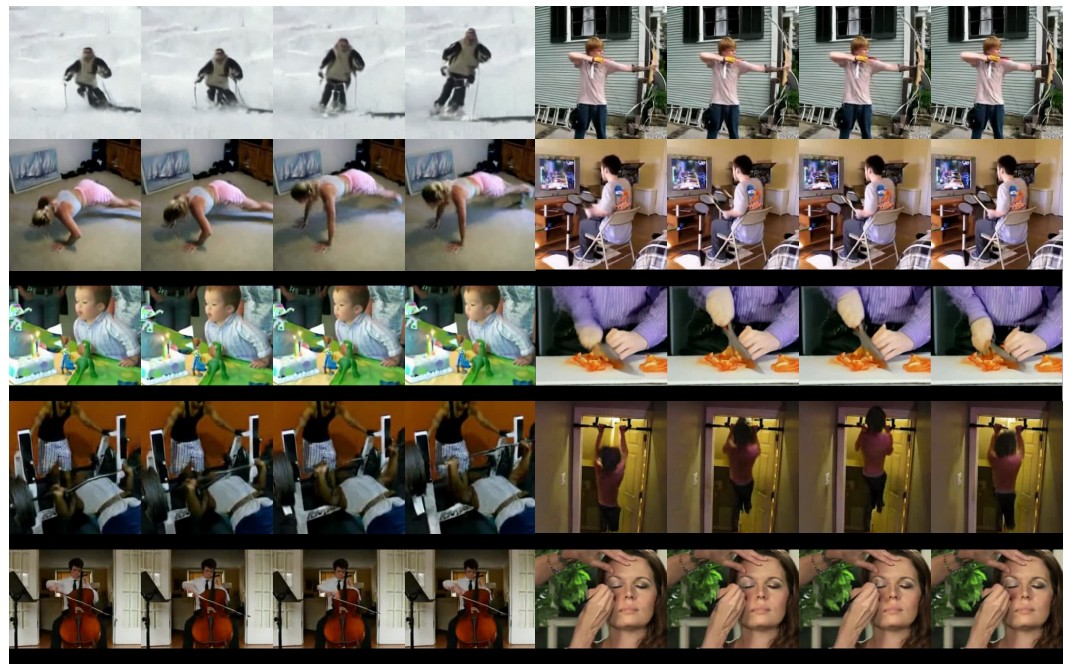

Figure 8: Qualitative results for video generation with CFG on the UCF101 dataset.

Table 3: Comparison of unconditional video generation on LAVID at 60 FPS.

| Method | #Tokens | TFLOPS↓ | gFVD↓ |
|---|---|---|---|
| Omni [33] | 10240 | 22.67 | 375.47 |
| Cosmos-L [1] | 10240 | 22.67 | 1552.04 |
| Cosmos-M [1] | 5120 | 7.95 | 1176.80 |
| Cosmos-S [1] | 1280 | 1.35 | 863.36 |
| VFRTok-L | 512 | 0.49 | **148.68** |

Table 4: Effectiveness of Partial RoPE.

| $\tau_{\mathrm{RoPE}}$ | Reconstruction | | | | Generation |
|---|---|---|---|---|---|
| | PSNR↑ | SSIM↑ | LPIPS↓ | rFVD↓ | gFVD↓ |
| 0 | 12.17 | 0.2204 | 0.7897 | 4652.25 | - |
| 0.25 | 29.24 | 0.8886 | 0.0545 | 18.38 | 199.86 |
| 0.5 | 30.87 | 0.9106 | 0.0451 | 16.33 | **147.41** |
| 0.75 | 30.91 | 0.9094 | 0.0437 | 16.17 | 174.44 |
| 1 | 30.85 | 0.9083 | 0.0441 | 16.84 | 208.42 |

VFRTok converges significantly faster; for example, VFRTok-L achieves a maximum convergence speed of $21.6\times$ that of OmniTokenizer [33]. Qualitative results of generation are shown in Figure 8.

We also provide the generation results on the LAVIB [27] 60 FPS dataset in Table 3. As shown, all baseline methods [1, 11, 33] require $2.5\times$ the tokens to represent videos with higher frame rates, resulting in a near-quadratic increase in computation cost. For example, for videos of the same duration, the computational cost of DiT based on OmniTokenizer [33] at 24 FPS is $5.82$, where as at 60 FPS it increases to $22.67$. In contrast, VFRTok maintains a constant token count regardless of frame rate, which not only improves efficiency but also enables significantly faster DiT convergence. As a result, VFRTok-L achieves the optimal gFVD [30] score among all compared methods.

## 4.4 Ablation Study

**Partial RoPE.** We perform an ablation study on Partial RoPE by training DiT [36] with varying Partial RoPE factors $\tau_{\mathrm{RoPE}} \in \{0, 0.25, 0.5, 0.75, 1\}$. As shown in Table 4, smaller $\tau_{\mathrm{RoPE}}$ fails to provide adequate positional guidance, resulting in poorer reconstruction and generation quality, while higher $\tau_{\mathrm{RoPE}}$ results in excessive positional bias (Figure 6), leading to suboptimal performance. Specifically, we find that when $\tau_{\mathrm{RoPE}} = 0$, ViT [2] entirely loses its capacity for positional encoding, rendering it incapable of reconstructing or generating videos. We adopt $\tau_{\mathrm{RoPE}} = 0.5$ as our default setting, as it achieves an optimal balances position priors and content modeling. The visualization of latent-to-patch attention map belonging to the position heads and content heads is illustrated in Figure 9. VFRTok effectively decouples the two patterns, and different samples in the content

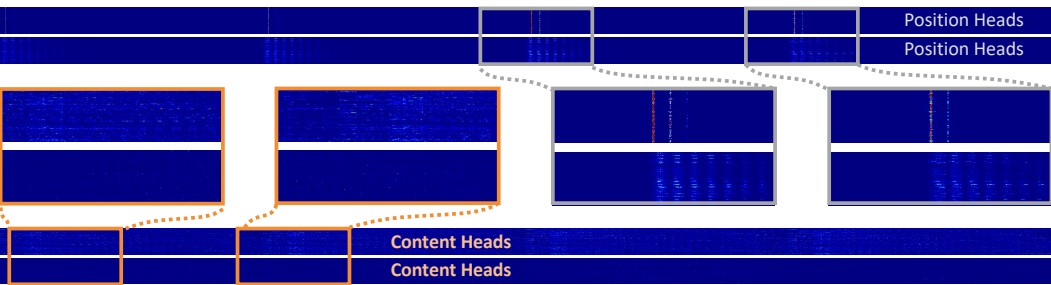

Figure 9: VFRTok decouples positional encoding and content modeling through Partial RoPE. The Position heads share similar patterns across samples (rows) indicating reduced sensitivity to content. The content heads attend to distinct patches in different samples (zoom in for details).

Table 5: Symmetric and asymmetric reconstruction performance on Adobe240fps dataset. *Symm* represents a variation of VFRTok which is trained on the symmetric reconstruction task.

| Methods | Symmetric Reconstruction (PSNR) | | | Asymmetric Reconstruction (PSNR) | | |
| | 30fps | 60fps | 120fps | $30 \rightarrow 60$fps | $30 \rightarrow 120$fps | $60 \rightarrow 120$fps |
|---|---|---|---|---|---|---|
| Symm | **25.92** | **24.71** | 23.89 | 22.36 | 21.60 | 23.33 |
| Ours | 25.87 ↓0.05 | 24.53 ↓0.18 | **23.95** ↑0.03 | **24.05** ↑1.69 | **23.56** ↑1.96 | **23.90** ↑0.66 |

head exhibit greater diversity in attention patterns. The PCA analysis in Figure 5(d) also shows that adopting partial RoPE can increase the variability of video regions influenced by individual tokens.

**Asymmetric vs. Symmetric Training.** We train a VFRTok variant under a symmetric encoding strategy. Specifically, we initialized from the K600 [3] pre-trained model in stage 1 for quick verification. In stage 2, which uses both K600 [3] and BVI-HFR [7] datasets, we disabled asymmetric encoding, forcing both encoder and decoder to operate at the same frame rate. For fair comparison, we adopt Adobe240fps [29] for evaluation. As shown in Table 5, this symmetric training yields a marginally better reconstruction in the symmetric setting but suffers a pronounced degradation under asymmetric encoding. Note that our stage 1 initialization itself used asymmetric frame rates, which partially narrows the gap in asymmetric reconstruction. Overall, these results confirm that asymmetric training is crucial for enabling VFRTok to generate videos at arbitrary frame rates.

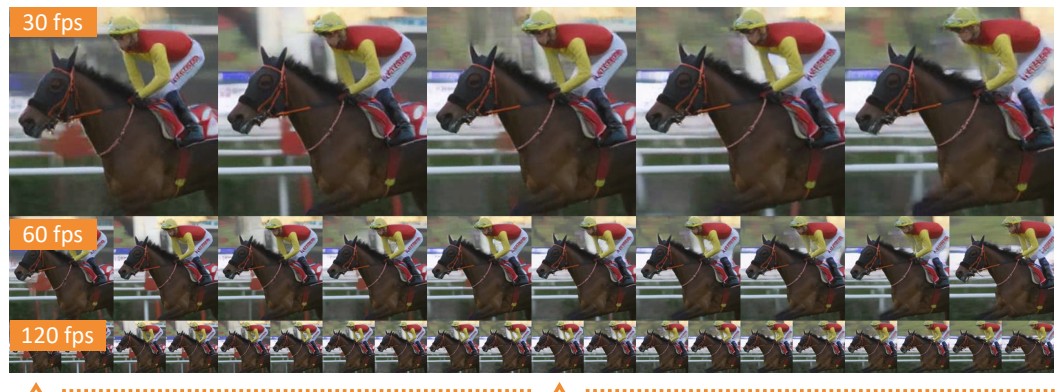

Figure 10: Results of video frame interpolation from 12 FPS to 30, 60, and 120 FPS. The original 12 FPS frame positions are indicated by triangle markers.

## 4.5 Video Frame Interpolation

VFRTok is designed for LDM, but also exhibits strong capabilities in Video Frame Interpolation (VFI). We evaluate VFRTok on sequences exhibiting large motion from the public UVG [23] dataset.

As shown in Figure 10, VFRTok successfully interpolates 12 FPS videos to 30, 60 and 120 FPS. A key advantage of VFRTok is its ability to perform variable frame rate interpolation, supporting arbitrary input and output frame rates. Performance on VFI can be enhanced by increasing the capacity of the latent space $\mathbb{R}^{N \times d}$ of VFRTok. Comparison with FLAVR [15] is shown in the Appendix B.

## 5  Conclusion

We propose the Duration-Proportional Information Assumption, where the upper bound on the observable information capacity of a video is proportional to the video duration. Under this assumption, we introduce VFRTok, a Transformer-based video tokenizer capable of encoding and decoding videos at variable frame rates. Furthermore, we introduce Partial Rotary Position Embeddings to decouple positional encoding from content modeling, thereby enhancing content-awareness, ultimately improving generation performance. Experiments show that VFRTok achieves comparable reconstruction performance and better generation quality compared to existing tokenizers using $1/8$ tokens, while being $11.9\times$ faster. Meanwhile, VFRTok converges significantly faster than existing works, reducing computational cost by up to $21.6\times$. When the frame rate increases, VFRTok does not require additional denoising tokens in DiT as frame rate increases, further demonstrating its efficiency advantage. Finally, we also find that VFRTok has the potential for video interpolation, capable of interpolating 12 FPS videos up to 120 FPS.

**Limitations.** Dense attention limits VFRTok's scalability to long videos. Segmenting videos into temporal slices with causal window attention is a potential solution that we leave for future work.

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

# A   Implementation Details

The detailed configuration of VFRTok and LightningDiT are shown in Table 6 and Table 7.

Table 6: Training configuration of VFRTok.

| Configuration | Value |
|---|---|
| image resolution | $256 \times 256$ |
| enc/dec hidden dimension | 768 |
| enc/dec #position heads | 6 |
| enc/dec #content heads | 6 |
| enc/dec #layers | 12 |
| enc/dec patch size | $4 \times 8 \times 8$ |
| enc/dec positional embedding | 3D RoPE (video), 1D APE (latent) |
| optimizer | AdamW |
| weight decay | 1e-4 |
| optimizer momentum | $\beta_1, \beta_2 = 0.9, 0.95$ |
| global batch size | 512 (stage1), 64 (stage2), 16 (stage3) |
| training steps | 30k (stage1), 200K (stage2), 100K (stage3) |
| base learning rate | 1e-4 (stage1 & stage2), 1e-5 (stage2) |
| learning rate schedule | cosine |
| augmentation | horizontal flip, center crop |
| perceptual weight $\lambda_1$ | 1 |
| discriminator | DINOv2-S |
| discriminator weight $\lambda_2$ | 0.2 |
| discriminator start | 30K |
| discriminator LeCAM | 0.001 |

Table 7: Training and inference configuration of LightningDiT-XL.

| Configuration | Value |
|---|---|
| hidden dimension | 1152 |
| #heads | 16 |
| #layer | 28 |
| patch size | 1 |
| positional embedding | APE |
| optimizer | AdamW |
| weight decay | 0 |
| optimizer momentum | $\beta_1, \beta_2 = 0.9, 0.95$ |
| global batch size | 128 (UCF101/K600), 48 (LAVID) |
| training steps | 100K (UCF101/K600), 50K (LAVID) |
| base learning rate | 1e-4 |
| learning rate schedule | constant |
| augmentation | center crop |
| diffusion sampler | Euler |
| diffusion steps | 50 |
| CFG interval start | 0.1 |
| timestamp shift | 2 |

**Learnable token details.** We adopted the same design as existing image 1D tokenizers: on the encoder side, we learn $N$ independent latent tokens, whereas on the decoder side, we use a single shared token. Although we experimented with replacing the decoder's shared token with a fixed-length set of independent tokens, which improves reconstruction fidelity, this change eliminated the decoder's ability to flexibly handle variable-frame-rate decoding.

# B   Quantitative Results for Video Frame Interpolation

Table 8 shows quantitative video-interpolation results comparing FLAVR [15] and VFRTok. Although FLAVR is a strong video-interpolation baseline and outperforms VFRTok on this task, VFRTok was not designed primarily for interpolation. First, VFRTok employs an extremely high compression rate to enable efficient video generation, creating a tighter bottleneck than dedicated interpolation models. Second, our training set contains only 22 clips at 120fps and no 60fps videos, whereas interpolation models are typically trained on large-scale, high-frame-rate data. In summary, while VFRTok can perform interpolation, its principal application remains general video generation.

Table 8: Quantitative results for video interpolation.

| | $12 \rightarrow 24$ (2×) | | $30 \rightarrow 60$ (2×) | | $12 \rightarrow 48$ (4×) | | $30 \rightarrow 120$ (4×) | | $15 \rightarrow 120$ (8×) | |
| | PSNR | SSIM | PSNR | SSIM | PSNR | SSIM | PSNR | SSIM | PSNR | SSIM |
|---|---|---|---|---|---|---|---|---|---|---|
| FLAVR | 26.05 | 0.7852 | 34.22 | 0.9529 | 22.17 | 0.6435 | 27.59 | 0.8372 | 21.70 | 0.6135 |
| Ours | 22.93 | 0.6724 | 24.06 | 0.7324 | 22.08 | 0.6433 | 23.68 | 0.7215 | 19.58 | 0.5279 |

