# OpenReview forum: "VFRTok: Variable Frame Rates Video Tokenizer with Duration-Proportional Information Assumption"
_NeurIPS.cc/2025/Conference — NeurIPS 2025 poster_

### Official Review · Reviewer_bssa · 2025-07-02

**Clarity:** 2
**Significance:** 2
**Originality:** 3
**Rating:** 4
**Confidence:** 4

**Summary:**

This paper proposes VFRTok, a video tokenizer based on the **duration-proportional information assumption**, aiming to address the inefficiency in current video generation models caused by variations in frame rate. Existing video tokenizers typically use a fixed compression rate, resulting in the computational cost of diffusion models scaling linearly with frame rate. VFRTok introduces a variable frame rate encoding/decoding mechanism and incorporates an improved RoPE (Partial RoPE) to achieve more compact and continuous spatiotemporal representations.

**Questions:**

1. To my knowledge, models such as WFVAE, Hunyuan VAE, and Wan VAE have been trained extensively across different frame rates and can also reconstruct and generate videos at arbitrary frame rates. What advantages does your approach offer compared to these methods?
2. Since there are multiple choices for frame rate, the number of frames corresponding to the same duration will vary. If the number of learnable tokens is fixed, it may be difficult to adaptively align the output frame count. How is training conducted in this case? Is the number of tokens constant across different frame rates? Does information loss occur when the frame rate is too high?
3. Is the Duration-Proportional Information Assumption applicable to all types of video?
4. What are the results of symmetric training across frame rates? This is not analyzed in the paper.
5. Partial RoPE applies RoPE only to some attention heads. While this improves content modeling, could it potentially weaken the model's ability to accurately model spatiotemporal positions, thereby affecting generation consistency?

**Ethical Concerns:**

["NO or VERY MINOR ethics concerns only"]

**Final Justification:**

The author has addressed part of my concerns. I will keep my score at Borderline Accept.

**Limitations:**

see Questions

**Paper Formatting Concerns:**

Terms such as "token," "latent token," and "latent representation" are used interchangeably. It is recommended to provide unified definitions and highlight them (e.g., in bold or italics) on first appearance.

**Quality:**

3

**Strengths And Weaknesses:**

**Strengths:**

1. The introduction of Partial RoPE cleverly addresses the issue of content modeling interference by positional modeling in Transformers.
2. The method supports video interpolation at arbitrary frame rates (from 12 FPS to 120 FPS), demonstrating strong generalization ability.
3. The paper is logically well-structured and contains rich visualizations and figures.

**Weaknesses:**

1. The paper does not discuss in depth how the Duration-Proportional Information Assumption theoretically guarantees the effectiveness of this compression method (e.g., information fidelity, reconstruction error).
2. Some experimental details (such as optimizer settings, learning rate schedules, regularization methods) are not fully described in the main text, affecting the reproducibility and independent evaluation of the work.

---

> ### Author Rebuttal · Authors · 2025-07-30
>
> We appreciate your positive feedback on the architecture design and paper logic.
> Below, we address each of the noted weaknesses and questions in turn.
>
> # W1/Q2 - Duration-Proportional Information Assumption and practice
>
> The Duration‑Proportional Information Assumption is motivated by three observations:
>
> - Video information content grows as frame rate increases.
> - However, the rate of that growth diminishes at higher frame rates.
> - As the frame rate approaches infinity, the total information converges to a finite upper bound. (Equation 1)
>
> The traditional tokenizers uses a fixed spatiotemporal compression rate, which implicitly assumes a linear, unbounded increase in information with frame rate.
> In contrast, VFRTok instead employs a fixed-length latent sequence to approximate that theoretical upper bound.
>
> In the table below, we compare reconstruction performance on the Adobe240fps dataset between standard tokenizers and VFRTok across multiple frame rates.
> As expected, the traditional tokenizers' token count and its reconstruction quality grows linearly with frame rate, implying that it overallocates capacity at high frame rates.
> While VFRTok's reconstruction quality degrades modestly at higher frame rates.
> The experimental result confirms that real video information scales sublinearly.
> We note that VFRTok was trained primarily on 24fps and 30fps data, with only 22 clips at 120fps; increasing the proportion of high–frame‑rate examples should further improve its fidelity at those rates.
>
> | FPS | Metrics | ｜Omni | Cosmos-L | Cosmos-M | **Cosmos-S** | **Ours** |
> | --- | --- | --- | --- | --- | --- | --- |
> | 24 | #Tokens | ｜4096 | 4096 | 2048 | **512** | **512** |
> | 24 | PSNR | ｜25.38 | 27.69 | 26.06 | **24.06** | **25.85** |
> | 60 | #Tokens | ｜10240 | 10240 | 5120 | **1280** | **512** |
> | 60 | PSNR | ｜25.64 | 29.16 | 27.81 | **25.47** | **24.53** |
> | 120 | #Tokens | ｜20480 | 20480 | 10240 | **2560** | **512** |
> | 120 | PSNR | ｜23.60 | 29.80 | 28.53 | **26.09** | **23.95** |
>
> While neither the Frame‑Proportional nor the Duration‑Proportional Information Assumption perfectly captures the sublinear growth of video information with frame rate, the Duration‑Proportional Assumption nonetheless offers clear practical advantages over a purely Frame‑Proportional approach in downstream generation.
> By fixing the latent length, it enables a single DiT model to denoise and generate videos at arbitrary frame rates, and it delivers better efficiency gains as target frame rate increases.
>
> # W2 - Experimental details
>
> We will add more experimental details in the appendix in the next version.
> The configuration of VFRTok and LightningDiT are shown in the following tables.
>
> | VFRTok Configuration | Value |
> | --- | --- |
> | image resolution | 256$\times$256 |
> | enc/dec hidden dimension | 768 |
> | enc/dec #position heads | 6 |
> | enc/dec #content heads | 6 |
> | enc/dec #layers | 12 |
> | enc/dec patch size | 4$\times$8$\times$8 |
> | enc/dec positional embedding | 3D RoPE (video), 1D APE (latent) |
> | optimizer | AdamW |
> | weight decay | 1e-4 |
> | optimizer momentum | $\beta_1,\beta_2=0.9,0.95$ |
> | global batch size | 128 (stage1), 32 (stage2) |
> | training steps | 200K (stage1), 100K (stage2) |
> | base learning rate  | 1e-4 (stage1), 1e-5 (stage2) |
> | learning rate schedule | cosine |
> | augmentation | horizontal flip, center crop |
> | perceptual weight $\lambda_1$ | 1 |
> | discriminator | DINOv2-S |
> | discriminator weight $\lambda_2$ | 0.2 |
> | discriminator start | 30K |
> | discriminator LeCAM | 1e-3 |
>
> | LightningDiT Configuration | Value |
> | --- | --- |
> | hidden dimension | 1152 |
> | #heads | 16 |
> | #layer | 28 |
> | patch size | 1 |
> | positional embedding | APE |
> | optimizer | AdamW |
> | weight decay | 0 |
> | optimizer momentum | $\beta_1,\beta_2=0.9,0.95$ |
> | global batch size | 128 (UCF101/K600), 48 (LAVID) |
> | training steps | 100K (UCF101/K600), 50K (LAVID) |
> | base learning rate | 1e-4 |
> | learning rate schedule | constant |
> | augmentation | center crop |
> | diffusion sampler | Euler |
> | diffusion steps | 50 |
> | CFG interval start | 0.1 |
> | timestamp shift | 2 |
>
> # Q1 - Advantages compared to existing works
>
> Although prior methods also train on raw videos at multiple frame rates, their strategies differ fundamentally from VFRTok.
> They either resample every video to a single, fixed frame rate before encoding or simply ignore frame rate by feeding all frames into the model.
> The former approach constrains the model to that chosen frame rate.
> While the latter enforces a uniform spatiotemporal compression rate regardless of frame rate, forcing high-frame-rate videos to be “slowed down” into unnecessarily many tokens and low-frame-rate videos to be “speed up”.
> Both of which limit the expressivity of the latent representation.
>
> As shown in Figure 1, these methods operate under a Frame‑Proportional Information Assumption, where token count grows linearly with frame rate (*e.g.*, a 120 fps clip demands four times the tokens of a 30 fps clip).
> By contrast, we find that a video’s information content increases sublinearly with frame rate and asymptotically approaches an upper bound (Duration-Proportional Information Assumption).
> VFRTok leverages this insight by using a fixed number of latent tokens to approximate that bound, thus unifying videos of the same duration but different frame rates under one consistent representation.
> Since the downstream DiT model’s computational complexity scales geometrically with the number of latent tokens, VFRTok’s constant-length encoding delivers significant efficiency gains for high-frame-rate video generation.
>
> # Q3 - Applicability
>
> When contrasting the Duration‑Proportional Information Assumption with the Frame‑Proportional Information Assumption, we are claiming that allocating information uniformly over time yields statistically better performance than allocating it uniformly over frames.
> However, at the sample level, both assumptions share a common limitation: videos exhibit widely varying content complexity.
> Specifically, videos vary in semantic, textural, and motion complexity, and thus should require varying information capacity.
> In conventional video codecs (*e.g.*, H.264), achieving the same reconstruction quality for two videos of differing complexity typically demands different bitrates.
> Analogously for tokenizers, the length of the latent representation ought to adapt to a video’s content complexity.
> This content‑adaptive modeling perspective is orthogonal to the hypothesis explored in our paper.
>
> While content‑adaptive tokenizers have been explored in a few works, the key challenge lies in enabling the downstream DiT model to support estimating a video’s complexity from its caption alone.
> We plan to address this in future work.
>
> # Q4 - Symmetric training
>
> We train a VFRTok variant under a symmetric encoding regime.
> Specifically, we initialized from the K600 pre-trained model in stage 1 for quick verification.
> In stage 2, which uses both K600 and BVI-HFR datasets, we disabled asymmetric encoding, forcing both encoder and decoder to operate at the same frame rate.
> As shown in the following table, this symmetric training yields a marginally better reconstruction in the symmetric setting but suffers a pronounced degradation under asymmetric encoding.
> Note that our stage 1 initialization itself used asymmetric frame rates, which partially narrows the gap in asymmetric reconstruction.
> Overall, these results confirm that asymmetric training is crucial for enabling VFRTok to generate videos at arbitrary frame rates.
> | **PSNR** | **30fps** | **60fps** | **120fps** | ｜**PSNR** | **30→60fps** | **30→120fps** | **60→120fps** |
> | --- | --- | --- | --- | --- | --- | --- | --- |
> | **Symm** | **25.92** | **24.71** | 23.89 | ｜**Symm** | 22.36 | 21.60 | 23.33 |
> | **Ours** | 25.87 $\downarrow0.05$ | 24.53$\downarrow0.18$ | **23.95**$\uparrow0.03$ | ｜**Ours** | **24.05**$\uparrow1.69$ | **23.56**$\uparrow1.96$ | **23.90**$\uparrow0.66$ |
>
> # Q5 - Partial RoPE and generation consistency
>
> Our experiments show that adding Partial RoPE does not compromise temporal consistency.
> As reported in the following table, we compute the average L2 distance between consecutive frames with and without Partial RoPE.
> The results reveal only a negligible difference, confirming that Partial RoPE maintains temporal coherence.
>
> Meanwhile, we find that incorporating Partial RoPE yields markedly more coherent motion in the generated videos.
> In particular, with it enabled, objects and human figures exhibit substantially less fragmentation, deformation, and distortion throughout their movements.
>
> | L2 distance | K600 | UCF101 |
> | --- | --- | --- |
> | w/o Partial RoPE | 13.93 | 12.26 |
> | w/ Partial RoPE | 14.14 | 11.77 |

---

> > ### Comment · Reviewer_bssa · 2025-08-04
> > **DINO**
> >
> > Thank you for your response.
> >
> > I noticed that you use DINO as the discriminator in your configuration, which differs from Hunyuan-VAE and Wan-VAE. Is using DINO as a discriminator a better approach? Is there any research that supports this?
> >
> > Additionally, I would suggest adding a discussion comparing your work with Hunyuan-VAE and Wan-VAE in the "Related Work" section.
> >
> > I will increase the score for a clear response to these points.

---

> > > ### Author Response · Authors · 2025-08-04
> > >
> > > # Q1 -  DINO discriminator
> > >
> > > Leverage a pre-trained DINOv2 model as a discriminator is a common choice supported by prior work, which demonstrates that ViT features from unsupervised DINO training capture rich global semantics [R1] and elevate generator quality [R2]. Additionally, VFRTok follows the configuration of image 1D tokenizers [5,6] to simplify the design.
> > >
> > > Notably, Wan-VAE employs an undisclosed 3D discriminator, while Hunyuan-VAE has yet to reveal its discriminator details. Inspired by these approaches, we believe that integrating a 3D discriminator with VFRTok could further enhance the model’s temporal consistency.
> > >
> > > [R1] *Li X, Qiu K, Chen H, et al. Xq-gan: An open-source image tokenization framework for autoregressive generation[J]. arXiv preprint arXiv:2412.01762, 2024.*
> > >
> > > [R2] *Li X, Qiu K, Chen H, et al. Imagefolder: Autoregressive image generation with folded tokens[J]. arXiv preprint arXiv:2410.01756, 2024.*
> > >
> > > # Q2 - Compare with Hunyuan- and Wan-VAE
> > >
> > > Thank you for the suggestion. In the camera-ready version, we will include detailed references to and comparisons with Hunyuan-VAE and Wan-VAE. Both Hunyuan-VAE and Wan-VAE adopt the Frame-Proportional Information Assumption, similar to Cosmos [1], OmniTokenizer [26] and LTX-VAE [10]. However, they outperform the open-source methods [1, 10, 26] primarily because they are trained on larger, proprietary datasets, and benefit from more refined parameter tuning.

---

### Official Review · Reviewer_VrQd · 2025-07-03

**Clarity:** 2
**Significance:** 4
**Originality:** 3
**Rating:** 5
**Confidence:** 4

**Summary:**

This paper introduces VFRTok, a query-based 1D transformer tokenizer that encodes videos of variable frame rate and same length into the same number of latent tokens. This is achieved by converting the frame index to the timestamp when computing the temporal rotation angles of a 3D RoPE. They further propose Partial RoPE, where they only apply RoPE to a subset of the attention heads. The authors conduct experiments on video reconstruction and generation based on VFRTok.

**Questions:**

- Sec. 4.5 shows that VFRTok can perform frame interpolation. Given a variable frame rate video LDM, can VFRTok be used to speed up inference by letting the LDM to generate videos latents with lower FPS and then use VFRTok to interpolate to higher FPS during or after decoding? If this is possible, this will be a significant contribution to the video generation field.
- Code is provided in supplementary material. Will the code and model weights be released to the public?
- From Fig. 3, it seems that only the patch tokens have RoPE while the latent tokens don't. Why such design and has this been ablated?

**Ethical Concerns:**

["NO or VERY MINOR ethics concerns only"]

**Final Justification:**

My questions are addressed in the rebuttal. I only have a minor suggestion on improving the clarity of figures and captions (see comment below). I maintain my rating of accept, since the method is a novel contribution and should have high impact in the field, where the number of tokens is a key bottleneck in the training and inference efficiency of video DiTs.

**Limitations:**

Yes.

**Quality:**

3

**Strengths And Weaknesses:**

## Strengths
- VFRTok achieves competitive reconstruction results and sota generation results while converge much faster.
- Both variable frame rate design and Partial RoPE are novel contributions to video tokenization and easy to implement.
- Partial RoPE is motivated by empirical observation where the attention is dominated by the grid position prior in Fig. 6.
- Partial RoPE is empirically validated in ablation study in Tab. 4.
- VFRTok supports interpolating videos from 12 FPS to 120 FPS.

## Weaknesses
- Some figures (e.g. Fig. 1-6), while informative, are a bit confusing and needs additional captions or paragraphs in the main paper.
- More details for the training (expanding L138-L145) and inference are needed. For example, how VFRTok adjust encoding/decoding frame rates during inference? Do VFRTok learn shared or separate sets of learnable tokens for each frame rate?
- No quantitative results for video interploation in Sec. 4.5.

---

> ### Author Rebuttal · Authors · 2025-07-30
>
> We appreciate your positive feedback on the technical novelty and experimental results.
> Below, we address each of the noted weaknesses and questions in turn.
>
> # W1 - Additional captions
>
> We will further refine captions to enhance readability.
> For example, we may modify the caption of Figure 1 to “*The number of tokens for other tokenizers grows with frame rate. VFRTok maintains a fixed number of latent tokens tied to video duration and supports asymmetric frame‑rate encoding and decoding.”*
>
> # W2 - Experimental details
>
> We will expand on the details of the training.
>
> **Inference details.** During inference, VFRTok supports decoding videos of any frame rate from latent tokens denoised by DiT.
> Specifically, the VFRTok decoder calculates the number of learnable tokens based on the incoming frame rate and injects the timestamp-based RoPE into each token.
>
> **Learnable token details.** We adopted the same design as existing image 1D tokenizers: on the encoder side, we learn N independent latent tokens, whereas on the decoder side, we use a single shared token.
> Although we experimented with replacing the decoder’s shared token with a fixed-length set of independent tokens, which improves reconstruction fidelity, this change eliminated the decoder’s ability to flexibly handle variable‑frame‑rate decoding.
> # W3 - Quantitative results for video interpolation
>
> The following table shows quantitative video‑interpolation results comparing FLAVR and VFRTok.
> Although FLAVR is a strong video‑interpolation baseline and outperforms VFRTok on this task, VFRTok was not designed primarily for interpolation.
> First, VFRTok employs an extremely high compression rate to enable efficient video generation, creating a tighter bottleneck than dedicated interpolation models.
> Second, our training set contains only 22 clips at 120fps and no 60fps videos, whereas interpolation models are typically trained on large-scale, high‑frame‑rate data.
> In summary, while VFRTok can perform interpolation, its principal application remains general video generation.
>
> | **PSNR** | **12→24** | **30→60** | **12→48** | **30→120** | **15→120** |｜ **SSIM** | **12→24** | **30→60** | **12→48** | **30→120** | **15→120** |
> | --- | --- | --- | --- | --- | --- | --- | --- | --- | --- | --- | --- |
> | **FLAVR** | 26.05 | 34.22 | 22.17 | 27.59 | 21.70 | ｜**FLAVR** | 0.7852 | 0.9529 | 0.6435 | 0.8372 | 0.6135 |
> | **Ours** | 22.93 | 24.06 | 22.08 | 23.68 | 19.58 | ｜**Ours** | 0.6724 | 0.7324 | 0.6433 | 0.7215 | 0.5279 |
>
> # Q1 - High frame rate video generation
>
> Traditional Frame-Proportional Information Assumption tokenizers must encode high frame rate videos during DiT training to enable high frame rate generation.
> In contrast, VFRTok’s Duration‑Proportional Information Assumption allows it to map videos of any frame rate into fixed‑length latent sequences.
> Because the VFRTok decoder converts these fixed‑length latents into variable‑length patch tokens, it can efficiently generate high frame rate videos, even when the DiT model is trained exclusively on low frame rate data.
>
> # Q2 - Release model weights
>
> We plan to release both the code and model weights concurrent with the camera‑ready version.
>
> # Q3 - Position embedding for latent tokens
>
> We adopt the architecture of existing 1D image tokenizers by incorporating learnable absolute positional encodings into our latent tokens.
> We omit RoPE because modeling relative positions between latent tokens is unnecessary.
> Unlike video patch tokens, where proximity in time and space implies similar content, the latent tokens aggregate global information, so adjacent latents need not correspond to semantically related features.

---

> > ### Comment · Reviewer_VrQd · 2025-08-08
> >
> > Thanks for the detailed rebuttal. Most of my concerns are addressed. I maintain my score.
> >
> > For the revision, I suggest improve the figure captions so that the figure is self-contained and won't cause any misunderstandings or confusion. For example, in figure 1, the caption should describe what the arrows represent, speficially for the 2-branched arrows and the dotted arrow. Also, I suggest include explanations and references where VFRTok inherits designs from existing 1d image tokenizers.

---

### Official Review · Reviewer_Zg58 · 2025-07-03

**Clarity:** 3
**Significance:** 3
**Originality:** 3
**Rating:** 4
**Confidence:** 4

**Summary:**

This paper proposes a new video tokenizer for video generation. The method is based on the assumption that a video's information is upper bounded by its duration not the number of frames. The tokenizer is based on recent 1D tokenizers using Transformers that directly patchify pixel cubes into tokens and produce latent tokens with learnable queries. On top of the 1D tokenizer design, the authors introduced 1) the split-head 3D RoPE which was previously used in Flux and QWen models; 2) the position heads and content heads, where the content heads are similar to NoPE. The tokenizer is trained with random pairs of high-to-low or low-to-high FPS rates in the inputs and outputs, effectively learning frame interpolation or sumsampling.

The proposed tokenizer is tested in video reconstruction and video generation benchmarks, showing better compression ratio and generation quality than grid based video tokenizers, while using few number of latent tokens.

**Questions:**

My major question relates to weakness #2: how to deal with varying spatial resolution with the current tokenizer design?

**Ethical Concerns:**

["NO or VERY MINOR ethics concerns only"]

**Final Justification:**

No other concern raised. Keeping my initial recommendation.

**Limitations:**

Yes.

**Quality:**

4

**Strengths And Weaknesses:**

+ Making video tokenizer less dependent on the frame number, which is an artifact of the video production process, seems natural and beneficial for video tokenization. The choice of Transformer-based 1D tokenizer makes this viable.
+ The design choices 1) and 2) seem to be sound and reasonable.
+ The training strategy that uses the frame interpolation/subsampling to make model adative to frame rates is very interesting and could generate good impact in the community.
+ Strong experimental results.

Weaknesses

- The 3D rope design and NoPE usage both lack references to the existing literature. I would recommend the authors to carefully revise the related work section.
- Video data has three dimensions in general, height, width, and time. In this work with a fixed latent token amount, the time dimension is flexible. But the choice of using 1D tokenizer with fixed latent token count makes it non-trivial to deal with the varying height and width of the video frames.

---

> ### Author Rebuttal · Authors · 2025-07-30
>
> We appreciate your positive evaluation of our work’s motivation, design strategy, and experimental findings.
> Below, we address each of the noted weaknesses and questions in turn.
>
> # W1 - RoPE related works
>
> We will include more comprehensive references to 3D RoPE and NoPE in the camera‑ready manuscript.
> It is important to note that existing 3D RoPE follows the same channel‑division strategy as VFRTok: each attention head’s channels are split into three groups, with RoPE applied independently along the temporal, width, and height dimensions (Equation 2).
> To our knowledge, no other work employs this split-head Partial RoPE approach.
>
> # W2/Q1 - Generalization to resolution
>
> Extending VFRTok to handle variable resolutions is another direction we are exploring.
> A natural extension of our Duration‑Proportional Information Assumption is a Spatial‑Temporal‑Proportional Information Assumption, which would allow videos of differing frame rates and resolutions to be encoded into fixed‑length latents.
> In practice, we would achieve this simply by applying interpolated RoPE along the spatial dimensions (height and width) for different‑resolution inputs.
>
> That said, stretching the video representation across three dimensions may lead to capacity inefficiencies in low‑frame‑rate and low‑resolution segments.
> To mitigate this, we plan to design a sublinear, variable‑compression‑rate video tokenizer in future work.

---

> > ### Comment · Reviewer_Zg58 · 2025-08-05
> >
> > Thanks for the response.
> >
> > On varying resolution question, I am more curious about the 1D tokens part. They are in fixed length; when increasing and decreasing resolution, it may create a capacity mismatch. It would be beneficial to discuss these questions.

---

> > > ### Author Response · Authors · 2025-08-05
> > >
> > > # Q1 - Variable-length 1D latents
> > >
> > > The latent‐token length in a 1D tokenizer can itself be made variable, as demonstrated by FlexTok [R1]. FlexTok employs an larger latent budget during training and randomly drops tail tokens, compelling the model to reconstruct using any prefix of latents. We are now generalizing this concept to support variable spatial resolution and frame rate, thereby offering more flexible compression rates.
> > >
> > > [R1] *Bachmann R, Allardice J, Mizrahi D, et al. FlexTok: Resampling Images into 1D Token Sequences of Flexible Length[C]//Forty-second International Conference on Machine Learning. 2025.*

---

### Official Review · Reviewer_WTga · 2025-07-03

**Clarity:** 3
**Significance:** 3
**Originality:** 3
**Rating:** 5
**Confidence:** 4

**Summary:**

This paper introduce VFRTok, a transformer-based video tokenizer that trained through asymmetric frame rate training between the encoder and decoder. The length of the token encoded is propotional to the video duration instead of number of frames. In addition, a Partial RoPE is proposed to allow the model's content-aware learning. Experiments demonstrate the high compression rate and good reconstruction/generation quality. VFRTok also shows the potential in video frame interpolation.

**Questions:**

1. Is it possible to show evidence that when used as the tokenizer of a video generation model, VFRTok will not limit the contents' movement in video generation compared with other video tokenizers?
2. Since the Partial RoPE can improve the model's content-awareness, will it facilitate the video generation model to learn the knowledge of movement? For example, does it enable the video generation model to generate videos that better align with physical rules or exhibit more realistic motion patterns?

**Ethical Concerns:**

["NO or VERY MINOR ethics concerns only"]

**Final Justification:**

The proposed method is novel and valuable for real-world applications. The Duration-Proportional Information Assumption is a new and valuable perspective for research in video tokenizer.

**Limitations:**

yes

**Quality:**

3

**Strengths And Weaknesses:**

Strengths:
1. The proposed VFRTok is a potential solution to the inconsistency of frame rate in video generation data in real world practice, especially In scenarios like game video generation, where the frames in the training data are rendered by computers and it is difficult to keep the frame rate stable during rendering.
2. The experiment result seems promising. The high compression rate is a critical advantage for downstream applications, such as its use as a tokenizer in video generation models, where efficiency and compactness are essential.
3. The design of Partial RoPE is reasonable and effective.
4. When used as a tokenizer for video generation models, VFRTok 's flexibility in handling varying frame rates has the potential to alleviate the burden on the generation model by sharing the responsibility of frame rate control. It may also help mitigate the negative impact of inconsistent frame rates in training data, thereby improving the learning process of the video generation model.

Weakness:
I suspect that the underlying assumption of duration-proportional information has inherent limitations. This assumption appears valid only when the content within the frames is not moving rapidly. However, in cases where the content is moving quickly, videos with higher frame rates inherently capture significantly more information than those with lower frame rates over the same duration. From this perspective, I am concerned that VFRTok may inadvertently constrain the representation of fast-moving content when applied to video generation tasks. This limitation could restrict the model's ability to accurately capture dynamic scenes, potentially impacting its generation quality in scenarios involving rapid motion.

---

> ### Author Rebuttal · Authors · 2025-07-30
>
> We appreciate your positive feedback on our motivation and architectural design.
> Thank you as well for illustrating the application scenario of game video generation.
> Below, we address each of the noted weaknesses and questions in turn.
>
> # W1 - Duration-Proportional Information Assumption and content-adaptive modeling
>
> We appreciate your concerns.
> It is true that fast‑moving videos often convey more information.
> However, it is important to note that high frame‑rate video and high‑speed motion are not synonymous: a video’s frame rate and the speed of the motion it depicts are two decoupled factors.
> Consequently, traditional video tokenizers, which hold the Frame-Proportional Information Assumption, face the same limitation.
> Given a fixed frame rate, a standard tokenizer allocates a fixed number of tokens to represent each frame (often far more than VFRTok).
> As motion increases, reconstruction quality naturally degrades under this scheme.
>
> More generally, videos vary in semantic, textural, and motion complexity, and thus should require varying information capacity.
> In conventional video codecs (*e.g.*, H.264), achieving the same reconstruction quality for two videos of differing complexity typically demands different bitrates.
> Analogously for tokenizers, the length of the latent representation ought to adapt to a video’s content complexity.
> This content‑adaptive modeling perspective is orthogonal to the hypothesis explored in our paper.
>
> While content‑adaptive tokenizers have been explored in a few works, the key challenge lies in enabling the downstream DiT model to support estimating a video’s complexity from its caption alone.
> We plan to address this in future work.
>
> # Q1 - Fast motion modeling
>
> VFRTok’s latent capacity is adequate for capturing rapid motion.
> As illustrated in Figure8, VFRTok successfully generates scenes of fast-paced activities, such as skiing and pull‑ups.
> Notably, the examples shown in Figure~8 are drawn from clips only 2–3 seconds in length, underscoring the model’s ability to represent and recreate high‑speed motion faithfully.
>
> # Q2: Partial RoPE enhances motion modeling
>
> Our analysis shows that incorporating Partial RoPE yields markedly more coherent motion.
> In particular, with Partial RoPE enabled, objects and human figures exhibit substantially less fragmentation, deformation, and distortion throughout their movements.

---

### Decision · Program_Chairs · 2025-09-17

**Decision:**

Accept (poster)

**Comment:**

The authors propose VFRTok, a query-based 1D transformer tokenizer that encodes videos of variable frame rate and same length into the same number of latent tokens. Additionally, the partial RoPE is proposed to encourage a functional separation between positional and content encoding. Promising results are demonstrated on video generation and video interpolation.

Initially, the reviewers raised several concerns, which are briefly outlined below:

* Reviewer WTga: Method's limitations.

* Reviewer Zg58: Lack of citing related works, limitations to deal with varying height and width of the video frames.

* Reviewer VrQd: Refinement on figures, details of training and inference, quantitative results for video interpolation.

* Reviewer bssa: Discussion on the Duration-Proportional Information Assumption, implementation details.

The rebuttal and subsequent author-reviewer discussions effectively addressed the reviewers' concerns. After carefully considering the reviews, rebuttal, and discussion, the AC concurs with the reviewers’ assessment (particularly, regarding VFRTok's potential to handle varying frame rates for video generation) and thus recommends acceptance of the paper.

Additionally, the AC kindly notes that the 1D image tokens approach [29] has also been extended to video [A] and further improved in subsequent work [B]. A proper discussion and citation of these related studies would strengthen the revision. Finally, the authors are encouraged to incorporate the rebuttal experiments into the manuscript and address the reviewers’ feedback in the final revision.


[A] LARP: Tokenizing Videos with a Learned Autoregressive Generative Prior. ICLR 2025.

[B] Democratizing Text-to-Image Masked Generative Models with Compact Text-Aware One-Dimensional Tokens. ICCV 2025